# DISCOVERING TOPICS WITH NEURAL TOPIC MODELS BUILT FROM PLSA LOSS

## ABSTRACT

In this paper we present a model for unsupervised topic discovery in texts corpora. The proposed model uses documents, words, and topics lookup table embedding as neural network model parameters to build probabilities of words given topics, and probabilities of topics given documents. These probabilities are used to recover by marginalization probabilities of words given documents. For very large corpora where the number of documents can be in the order of billions, using a neural auto-encoder based document embedding is more scalable then using a lookup table embedding as classically done. We thus extended the lookup based document embedding model to continuous auto-encoder based model. Our models are trained using probabilistic latent semantic analysis (PLSA) assumptions. We evaluated our models on six datasets with a rich variety of contents. Conducted experiments demonstrate that the proposed neural topic models are very effective in capturing relevant topics. Furthermore, considering perplexity metric, conducted evaluation benchmarks show that our topic models outperform latent Dirichlet allocation (LDA) model which is classically used to address topic discovery tasks.

## 1   INTRODUCTION

Nowadays, with the digital era, electronic text corpora are ubiquitous. These corpora can be company emails, news groups articles, online journal articles, Wikipedia articles, video metadata (titles, descriptions, tags). These corpora can be very large, thus requiring automatic analysis methods that are investigated by the researchers working on text content analysis (Collobert et al., 2011; Cambria & White, 2014). Investigated methods are about named entity recognition, text classification, etc (Nadeau & Sekine, 2007; S., 2002).

An important problem in text analysis is about structuring texts corpora around topics (Daud et al., 2010; Liu & Zhang, 2012). Developed tools would allow to summarize very large amount of text documents into a limited, human understandable, number of topics. In computer science many definitions of the concept of *topic* can be encountered. Two definitions are very popular. The first one defines a topic as an entity of a knowledge graph such as Freebase or Wikidata (Bollacker et al., 2008; Vrandečić & Krötzsch, 2014). The second one defines a topic as probability distribution over words of a given vocabulary (Hofmann, 2001; Blei et al., 2003). When topics are represented as knowledge graph entities, documents can be associated to identified concepts with very precise meaning. The main drawback is that knowledge graphs are in general composed of a very large number of entities. For example, in 2019, Wikidata counts about 40 million entities. Automatically identifying these entities requires building extreme classifiers trained with expensive labelled data (Puurula et al., 2014; Liu et al., 2017). When topics are defined as probability distribution over words of vocabulary, they can be identified using unsupervised methods that automatically extract them from text corpora. A precursor of such methods is the latent semantic analysis (LSA) model which is based on the word-document co-occurrence counts matrix factorization (Dumais, 1990). Since then, LSA has been extended to various probabilistic based models (Hofmann, 2001; Blei et al., 2003), and more recently to neural network based models (Salakhutdinov & Hinton, 2009; Larochelle & Lauly, 2012; Wan et al., 2011; Yao et al., 2017; Dieng et al., 2017).

In this paper, we propose a novel neural network based model to automatically, in an unsupervised fashion, discover topics in a text corpus. The first variation of the model is based on a neural

networks that uses as input or parameters documents, words, and topics discrete lookup table embedding to represent probabilities of words given documents, probabilities of words given topics, and probabilities of topics given documents. However because in a given corpus, the number of documents can be very large, discrete lookup table embedding explicitly associating to each document a given embedded vector can be unpractical. For example, for the case of online stores such as Amazon, or video platforms such as Dailymotion or Youtube, the number of documents are in the order of billions. To overcome this limitation, we propose a model that generates continuous document embedding using a neural auto-encoder (Kingma & Welling, 2013). Our neural topic models are trained using cross entropy loss exploiting probabilistic latent semantic analysis (PLSA) assumptions stating that given topics, words and documents can be considered independent.

The proposed models are evaluated on six datasets: KOS, NIPS, NYtimes, TwentyNewsGroup, Wikipedia English 2012, and Dailymotion English. The four first datasets are classically used to benchmark topic models based on bag-of-word representation (Dua & Graff, 2017). Wikipedia, and Dailymotion are large scale datasets counting about one million documents. These latter datasets are used to qualitatively assess how our models behave on large scale datasets. Conducted experiments demonstrate that the proposed models are effective in discovering latent topics. Furthermore, evaluation results show that our models achieve lower perplexity than latent Dirichlet allocation (LDA) trained on the same datasets.

The remainder of this paper is organized as follows. Section 2 discusses related work. Section 3 briefly presents principles of topics generation with PLSA. Section 4 presents the first version of the model we propose which is based on discrete topics, documents, and words embedding. Section 5 gives details about the second version of the model which is based on embedding documents using a continuous neural auto-encoder. Section 6 provides details about the experiments conducted to assess the effectiveness of the proposed models. Finally Section 7 derives conclusions and gives future research directions.

## 2    RELATED WORK

Unsupervised text analysis with methods related to latent semantic analysis (LSA) has a long research history. Latent semantic analysis takes a high dimensional text vector representation and apply linear dimensionality reduction methods such as singular value decomposition (SVD) to the word-document counts matrix (Dumais, 1990). The main drawback of LSA is related to it's lack of statistical foundations limiting the model interpretability.

Probabilistic latent semantic analysis (PLSA) was proposed by Hofmann (2001) to ground LSA on solid statistical foundations. PLSA is based on a well defined generative model for text generation based on the bag-of-words assumption. PLSA can be interpreted as a probabilistic matrix factorisation of the word-document counts matrix. Because PLSA model defines a probabilistic mixture model, it's parameters can be estimated using the classical Expectation-Maximization (EM) algorithm (Moon, 1996). PLSA has been exploited in many applications related to text modelling by Hofmann (2001), to collaborative filtering by Popescul et al. (2001), to web links analysis by Cohn & Hofmann (2001), and to visual scene classification by Quelhas et al. (2007).

The main drawback of PLSA is that it is a generative model of the training data. It does not apply to unseen data. To extend PLSA to unseen data, Blei et al. (2003) proposed the latent Dirichlet Allocation (LDA) which models documents via hidden Dirichlet random variables specifying probabilities on a lower dimensional hidden spaces. The distribution over words of an unseen document is a continuous mixture over document space and a discrete mixture over all possible topics. Modeling with LDA has been thoroughly investigated resulting in dynamic topic models to account for topics temporal dynamics by Blei & Lafferty (2006); Wang et al. (2008); Shalit et al. (2013); Varadarajan et al. (2013); Farrahi & Gatica-Perez (2014), hierarchical topic models to account topic hierarchical structures by Blei et al. (2004), and multi-lingual topic model to account for multi-lingual corpora by Boyd-Grabber & Blei (2009); Vulic et al. (2015), and supervised topic model to account for corpora composed by categorized documents (Blei & McAuliffe, 2008). Beside text modelling, LDA has been applied to discover people's socio-geographic routines from mobiles phones data by Farrahi & Gatica-Perez (2010; 2011; 2014), mining recurrent activities from long term videos logs by Varadarajan et al. (2013).

Learning a topic models based on LSA, PLSA or LDA requires considering jointly all words, documents, and topics. This is a strong limitation when the vocabulary and the number of documents are very large. For example, for PLSA or LDA, learning the model requires maintaining a large matrix containing the probability of a topics given words and documents (Hofmann, 2001; Blei et al., 2003). To overcome this limitation Hoffman et al. (2010) proposed online training of LDA models using stochastic variational inference.

Recently, with the rise of deep learning with neural networks that are trained using stochastic gradient descent on sample batches, novel topic models based on neural networks have been proposed. Salakhutdinov & Hinton (2009) proposed a two layer restricted Boltzmann machine (RBM) called the replicated-softmax to extract low level latent topics from a large collection of unstructured documents. The model is trained using the contrastive divergence formalism proposed by Carreira-Perpiñán & Hinton (2005). Benchmarking the model performance against LDA showed improvement in term on unseen documents' perplexity and accuracy on retrieval tasks. Larochelle & Lauly (2012) proposed a neural auto-regressive topic model inspired from the replicated softmax model but replacing the RBM model with neural auto-regressive distribution estimator (NADE) which is a generative model over vectors of binary observations (Larochelle & Murray, 2011). An advantage of the NADE over the RBM is that during training, unlike for the RBM, computing the data negative log-likelihood's gradient with respect to the model parameters does not requires Monte Carlo approximation. Srivastava et al. (2013) generalized the replicated softmax model proposed by Salakhutdinov & Hinton (2009) to deep RBM which has more representation power.

Cao et al. (2015) proposed neural topic model (NTM), and it's supervised extension (sNTM) where words and documents embedding are combined. This work goes beyond the bag-of-words representation by embedding word n-grams with word2vec embedding as proposed by Mikolov et al. (2013). Moody (2016) proposed the lda2vec, a model combining Dirichlet topic model as Blei et al. (2003)) and word embedding as Mikolov et al. (2013). The goal of lda2vec is to embed both words and documents in the same space in order to learn both representations simultaneously.

Other interesting works combine probabilistic topic models such as LDA, and neural network modelling (Wan et al., 2011; Yao et al., 2017; Dieng et al., 2017). Wan et al. (2011) proposed a hybrid model combining a neural network and a latent topic model. The neural network provides a lower dimensional embedding of the input data , while the topic model extracts further structure from the neural network output features. The proposed model was validated on computer vision tasks. Yao et al. (2017) proposed to integrate knowledge graph embedding into probabilistic topic modelling by using as observation for the probabilistic topic model document-level word counts and knowledge graph entities embedded into vector forms. Dieng et al. (2017) integrated to a recurrent neural network based language model global word semantic information extracted using a probabilistic topic model.

## 3 TOPIC MODELLING WITH PROBABILISTIC LATENT SEMANTIC ANALYSIS

Probabilistic latent semantic analysis (PLSA) proposed by Hofmann (2001) is based on the bag-of-words representation defined in the following.

### 3.1 BAG OF WORDS REPRESENTATION

The grounding assumption of the bag-of-word representation is, that for text content representation, only word occurrences matter. Word orders can be ignored without harm to understanding.

Let us assume available a corpus of documents $\mathcal{D} = \{\text{doc}_1, \text{doc}_2, ..., \text{doc}_i, ..., \text{doc}_I\}$. Every document is represented as the occurrence count of words of a given vocabulary $\mathcal{W} = \{\text{word}_1, \text{word}_2, ..., \text{word}_n, ..., \text{word}_N\}$. Let us denote by $c(\text{word}_n, \text{doc}_i)$ the occurrence count of the $n$'th vocabulary word into the $i$'th document. The normalized bag-of-words representation of the $i$'th document is given by the empirical word occurrences probabilities:

$$f_{ni} = \frac{c(\text{word}_n, \text{doc}_i)}{\sum_{m=1}^{N} c(\text{word}_m, \text{doc}_i)}, \ n = 1, ..., N. \tag{1}$$

With the bag-of-words assumption, $f_{ni}$, is an empirical approximation of the probability that $\text{word}_n$ appears in document $\text{doc}_i$ denoted $p(\text{word}_n|\text{doc}_i)$.

(a) Document embedding with a lookup table.

(b) Embedding with an auto-encoder.

(c) Probabilities of topics given document.

(d) Probabilities of words given topic.

Figure 1: Discrete and continuous neural topic model. Neural network biases are omitted for figure clarity.

## 3.2 PROBABILISTIC LATENT SEMANTIC ANALYSIS

Probabilistic latent semantic analysis (PLSA) modelling is based on the assumption that there is a set unobserved topics $\mathcal{T} = \{\text{top}_1, \text{top}_2, ..., \text{top}_K\}$ that explains occurrences of words in documents. Given topics, words and documents can be assumed independent. Thus, under the PLSA assumption the probability of the occurrence a word $\text{word}_n$ in a document $\text{doc}_i$ can be decomposed as:

$$p(\text{word}_n | \text{doc}_i) = \sum_{k=1}^{K} p(\text{word}_n | \text{top}_k) p(\text{top}_k | \text{doc}_i) \qquad (2)$$

Hofmann (2001) used the expectation maximization algorithm (Moon, 1996) to estimate probabilities of words given topics $p(\text{word}_n | \text{top}_k)$, and probabilities of topics given documents $p(\text{top}_k | \text{doc}_i)$.

It is important to note that PLSA, as well as LDA, are trained on the raw counts matrix $(c(\text{word}_m, \text{doc}_i))$ and not the normalized counts matrix $(f_{ni})$. The normalized counts matrix are used by the models we propose the following sections.

## 4 DISCRETE NEURAL TOPIC MODEL

The discrete neural topic model we propose is based on a neural network representation of probabilities involved in representing the occurrences of words in documents: $p(\text{word}_n | \text{doc}_i)$, $p(\text{word}_n | \text{top}_k)$, and $p(\text{top}_k | \text{doc}_i)$. These probabilities are parametrized by the documents, the words, and the topics discrete lookup table embeddings.

Lets us denote by $\mathbf{x}_i = (x_{di})_{d=1}^{D}$ a D-dimensional[1] embedded vector representing the $i$'th document $\text{doc}_i$. Similarly, we define $\mathbf{y}_n = (y_{dn})_{d=1}^{D}$, and $\mathbf{z}_k = (z_{dk})_{d=1}^{D}$ D-dimensional embedded vectors respectively representing word $\text{word}_n$ and topic $\text{top}_k$. Using these discrete lookup embeddings as parameters, the probability of words given documents can be written as:

$$p(\text{word}_n | \text{doc}_i) = \frac{\exp(\mathbf{y}_n^\intercal \mathbf{x}_i + b_n)}{\sum_{m=1}^{N} \exp(\mathbf{y}_m^\intercal \mathbf{x}_i + b_m)}. \qquad (3)$$

Similarly, probabilities of words given a topic are defined as:

$$p(\text{word}_n | \text{top}_k) = \frac{\exp(\mathbf{y}_n^\intercal \mathbf{z}_k + b_n)}{\sum_{m=1}^{N} \exp(\mathbf{y}_m^\intercal \mathbf{z}_k + b_m)}, \qquad (4)$$

---

[1] $\mathbf{x}_i$ is a column vector in $\mathbb{R}^D$ .

and the probability of a topic given a document is defined as

$$p(\text{top}_k|\text{doc}_i) = \frac{\exp(\mathbf{z}_k^\intercal \mathbf{x}_i + b_k)}{\sum_{l=1}^{K} \exp(\mathbf{z}_l^\intercal \mathbf{x}_i + b_l)} \tag{5}$$

In Equations 3, 4, 5, and in following equations, although different, all neural networks biases are denoted by $b$. We used this convention to avoid burdening the reader with too many notations.

Figure 1a, Figure 1b and 1c give schematic representation of the neural network representation of probabilities of words given documents, probabilities of words given topics, and topics given documents. It is worth noticing that, because of the form of the probability of words given documents (see Equation 3) which is based on scalar product between word and document vectors, the higher probability of a word occurring in a document, the closer it's vector $\mathbf{y}_n$ to the document's vector $\mathbf{x}_i$ will be. Similar analysis can be derived about the proximity of word and topic vectors, and topic and document vectors.

The probabilities of words given topics (4), and topics given documents (5) can be combined according to the PLSA assumptions (Equation 2) to recover probabilities of the words given documents as:

$$p(\text{word}_n|\text{doc}_i) = \sum_{k=1}^{K} \frac{\exp(\mathbf{y}_n^\intercal \mathbf{z}_k + b_n)}{\sum_{m=1}^{N} \exp(\mathbf{y}_m^\intercal \mathbf{z}_k + b_m)} \times \frac{\exp(\mathbf{z}_k^\intercal \mathbf{x}_i + b_k)}{\sum_{l=1}^{K} \exp(\mathbf{z}_l^\intercal \mathbf{x}_i + b_l)} \tag{6}$$

To train the model, we optimize, using stochastic gradient descent, according to a cross entropy loss, embedding and biases parameters so that probabilities of words given documents in Equations 3, and 6 match the empirical bag-of-words frequencies defined in Equation 1.

## 5    CONTINUOUS NEURAL TOPIC MODEL

The discrete neural topic model described in Section 4 has two main drawbacks. First, it only models training data, and can not be applied to unseen data. Second, it requires building an explicit vector representation $\mathbf{x}_i$ for every document $i = 1, 2, ..., I$. In practise the number of documents can be very large, possibly in the order of billions. A solution to these issues is to use continuous embeddings to represent document instead of discrete lookup table embeddings (Vincent et al., 2010).

Continuous document embeddings are built using a neural auto-encoder representation that maps input documents $doc_i$ represented by their word empirical frequencies $f_{ni}$ onto themselves through a $D$-dimensional bottleneck layer $\mathbf{x}_i = (x_{di})_{d=1}^{D}$ which is then taken as the document embedding. This is done as:

$$\sigma\left(\sum_{n=1}^{N} y_{dn} f_{ni} + b_{di}\right) = x_{di} \tag{7}$$

$$\frac{\exp(\sum_{d=1}^{D} \tilde{y}_{dn} x_{di} + \tilde{b}_{dn})}{\sum_{m=1}^{N} \exp(\sum_{d=1}^{D} \tilde{y}_{dm} x_{id} + \tilde{b}_{dm})} = f_{ni} \tag{8}$$

where $\sigma$ is the rectified linear (ReLU) unit activation function. Variables $\mathbf{y} = (y_{dn})$, and $\tilde{\mathbf{y}} = (\tilde{y}_{dn})$ are neural network parameters, and $\mathbf{y}$ is taken to be word embeddings.

Figure 1b gives a schematic visualization of the continuous document embedding model. Because of it's continuous embeddings, this model can encode an unlimited number of documents as far as embedding dimension $D$ is large enough.

Similarly then for the discrete topic model, documents, words, and topics vector representation $\mathbf{x}_i$ and $\mathbf{y}_n$, and topics vectors $\mathbf{z}_k$ are combined to compute probabilities of words given topics using Equation 4, probabilities of topics given documents using Equation 5, and probabilities of words given documents using Equation 6.

To train the continuous neural topic model we optimized, using stochastic gradient descent, with respect to a cross entropy loss, parameters $\mathbf{x}_i$, $\mathbf{y}_n$ and $\mathbf{z}_k$ such that the models in Equations 7 and 6 match the empirical bag-of-words frequencies in Equation 1.

Table 1: Evaluation corpora statistics: $I$ is the corpus size, and $N$ the vocabulary size.

| Datasets | NIPS | KOS | NYTimes | 20NewsGroup | Wikipedia | Dailymotion |
|---|---|---|---|---|---|---|
| $I$ | 1500 | 3430 | 300000 | 18282 | 900000 | 1500000 |
| $N$ | 12419 | 6906 | 102660 | 24164 | 50000 | 50000 |

It has to be noticed, apart from biases, our models parameters are constituted by the embedding parameters. This allows to build a model with a limited set of parameters, exploiting parameters sharing as regularization procedure. For the auto-encoder model in Equations 7 we chose different encoding ($\mathbf{y}$) and decoding ($\tilde{\mathbf{y}}$) parameters to avoid over-constraining the model. However, if further reduction of the model number of parameters is targeted, these two variables can be considered as the transposed of one another.

# 6 EXPERIMENTS

## 6.1 EVALUATION PROTOCOL

We evaluated our models on six datasets. Four of them are classical dataset used to evaluate bag-of-words models: NIPS full papers, KOS blog entries, NYTimes news articles, and the Twenty News Group dataset. The three first datasets can be obtained from the UCI machine learning repository created by Dua & Graff (2017). The Twenty News Group dataset is part of datasets available with the well kown Python Scikit-Learn package.

The two other datasets are the Wikipedia English 2012, and the Dailymotion English used to assess qualitatively how our models perform on datasets with very large number of documents. Apart from the Dailymotion dataset, all other ones are publicly available, and can be used for model benchmarking. Table 1 gives the corpora's statistics. These corpora are very diverse in term of corpus sizes, vocabulary sizes, and document contents.

We evaluate the discrete neural topic model (D-NTM) presented in Section 4, and it's continuous extension (C-NTM) presented in Section 5. These models are compared to the latent Dirichlet allocation (LDA) model, developed by Blei et al. (2003), taken as baseline. We considered this baseline as it outperforms the PLSA models. We used the LDA implementation available in the Python Scikit-Learn package based on Hoffman et al. (2010) implementation.

To assess the performances of the models, as proposed by Hofmann (2001); Blei et al. (2003), we use the perplexity measure defined as:

$$pp = \exp -\frac{\sum_{i=1}^{I} \log p(\text{word}_1, ..., \text{word}_n, ..., \text{word}_{N_i}, \text{doc}_i)}{\sum_{d=1}^{M} N_i} \qquad (9)$$

where $\text{word}_1, ..., \text{word}_n, ..., \text{word}_{N_i}$ is the sequence of possibly duplicated words composing the $i$'th document $\text{doc}_i$, and :

$$p(\text{word}_1, ..., \text{word}_n, ..., \text{word}_{N_i}, \text{doc}_i) = \prod_{n=1}^{N_i} \sum_{k=1}^{K} p(\text{word}_n|\text{top}_k)p(\text{top}_k|\text{doc}_i)p(\text{doc}_i) \qquad (10)$$

The perplexity represents the exponential of data negative log-likelihood of estimated models. Thus, the smaller the better. This measure is classically used to assess language models, and topic models performances.

Our models comprise two hyper-parameters: the embedding dimension $D$, and the number of topics $K$. As we are optimizing our models using stochastic gradient descent, their training involves three parameters: a learning rate set to $\lambda = 0.01$, a number of descent steps set to $= 100$, and a batch size that was set to $64$. Our models were implemented in the Tensorflow framework. Neural network parameters were initialized with Xavier initializers, and model optimization is performed with Adam Optimizers [2].

---

[2] Our model implementation and benchmarking scripts will be made available upon conference reviews.

Table 2: Topic discovery benchmarking results. $K$ is the number of topics. For D-NTM and C-NTM results are presented for embedding dimensions of $D = 100, 200, 300$.

| Dataset | $K$ | D-NTM | | | C-NTM | | | LDA |
|---|---|---|---|---|---|---|---|---|
| | | 100 | 200 | 300 | 100 | 200 | 300 | |
| KOS | 50 | 1785.2 | 1786.1 | 1811.5 | 1957.3 | 1922.9 | 1893.4 | 2232.9 |
| | 100 | 1601.7 | 1546.2 | 1514.5 | 1708.8 | 1606.4 | 1583.8 | 2197.6 |
| | 200 | 1421.5 | 1283.7 | 1256.1 | 1511.5 | 1356.2 | 1292.7 | 2123.0 |
| | 300 | 1322.2 | 1159.8 | 1112.5 | 1434.7 | 1246.0 | 1160.2 | 2126.6 |
| NIPS | 50 | 3849.8 | 3902.3 | 3896.8 | 4328.1 | 4357.3 | 4428.8 | 4331.6 |
| | 100 | 3713.9 | 3623.4 | 3634.7 | 4048.7 | 3964.5 | 3926.1 | 4317.5 |
| | 200 | 3537.9 | 3357.1 | 3317.8 | 3747.8 | 3588.7 | 3574.5 | 4316.1 |
| | 300 | 3458.2 | 3247.4 | 3086.9 | 3631.2 | 3520.0 | 3321.2 | 4302.3 |
| 20NewsGroup | 50 | 4635.1 | 4611.1 | 4692.0 | 4646.0 | 4609.9 | 4494.9 | 4793.0 |
| | 100 | 3989.3 | 3857.9 | 3849.1 | 4067.3 | 3815.5 | 3721.2 | 4511.7 |
| | 200 | 3544.9 | 3273.0 | 3193.3 | 3577.5 | 3214.8 | 3023.1 | 4367.2 |
| | 300 | 3258.7 | 2964.3 | 2813.4 | 3304.2 | 2858.2 | 2739.9 | 4342.5 |

## 6.2 RESULTS

We investigate neural topic models training performances for varying embedding dimension $D$ and number of topics $K$. We tested number of topics of $K = 50, 100, 200, 300$, and embedding dimensions of $D = 100, 200, 300$.

Table 2 gives training perplexity for the models on the KOS, the NIPS, and the Twenty News Group datasets. These results show that the training perplexity decreases with the number of topics until a number where it stagnates. Also, the training perplexity is higher when the embedding dimension is about a 100, while for 200 and 300, exhibit close perplexity values. This trend with perplexity decreasing with increasing embedding dimension and number of topics is expected as larger dimension implies higher neural network learning capacity.

Table 2 also gives the comparison of training perplexity between D-NTM, C-NTM, and LDA. These results show that training perplexity is much lower for neural network based topic model than for LDA. They also show that, in general, D-NTM is more efficient at achieving low perplexity than C-NTM. For an embedding dimension $D = 300$, for the KOS and the NIPS dataset, the D-NTM model achieves better performances, while for the Twenty News Group, the C-NTM achieves better performances.

Figure 2 gives samples topics discovered using the continuous neural topic model (C-NTM) over large scale dataset: NYtimes, Wikipedia 2012, and Dailymotion. We only considered this model as it scales better than the discrete neural topic model (D-NTM) to large scale datasets. Discovered topics are displayed in form of word clouds where the size of each word is proportional to the probability the words occurs in the considered topic $p(\text{word}_n | \text{top}_k)$. This figure shows that the model find relevant topics. For NYtimes, the discovered topics are about energy plants, medecine, and court law. For Wikipedia displayed topics are about books and novels, universities and schools, and new species. For Dailymotion, discovered topics are about movies, videos productions, and Super Bowl. These qualitative results show that found topics are consistent and centered around concepts a human being can identify and expect. These examples are just few sample topics, other non displayed topics are about news, sport, music, religion, science, economy, etc.

## 7 CONCLUSIONS

In this paper we presented a novel neural topic model. The proposed model has two variations. The first variation is based on discrete documents, words, and topics discrete lookup table embeddings. The second variation exploits continuous neural auto-encoder embedding to allow scaling to very large corpora. Proposed models are evaluated on six datasets. Conducted evaluation demonstrate that proposed models outperform LDA with respect to a perplexity metric.

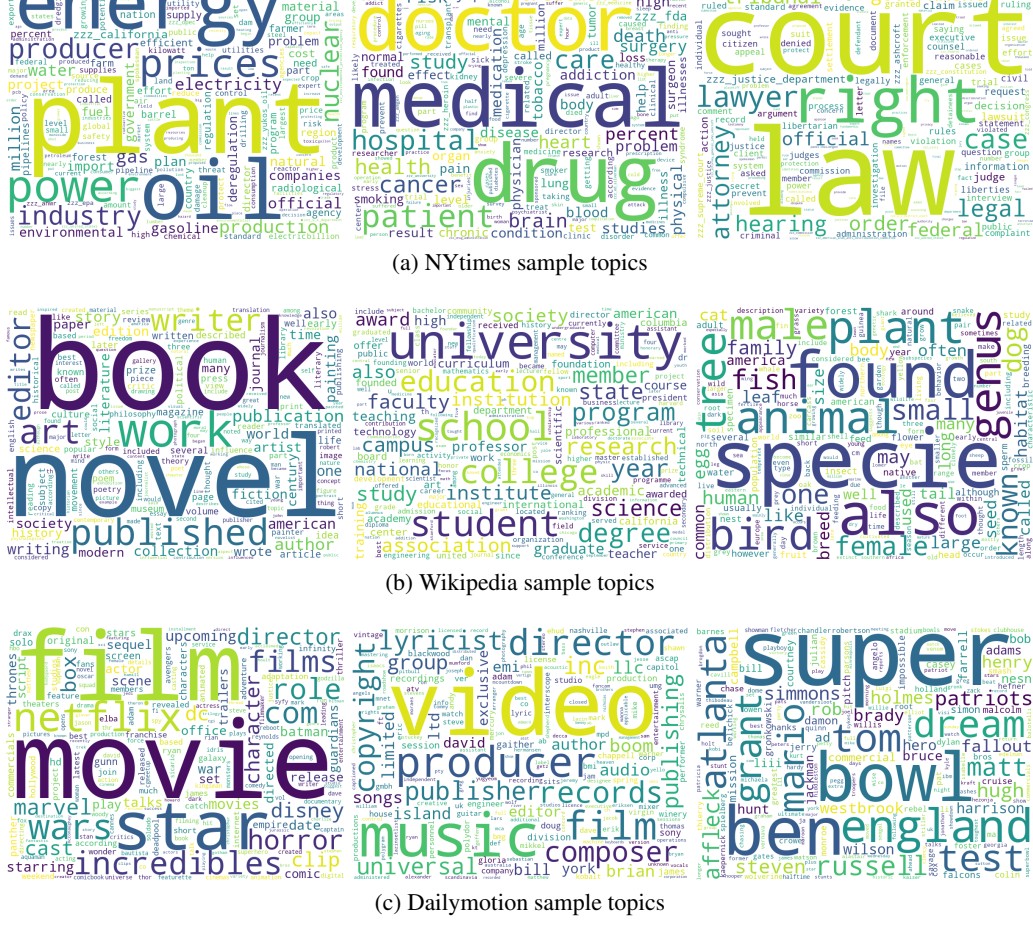

(a) NYtimes sample topics

(b) Wikipedia sample topics

(c) Dailymotion sample topics

Figure 2: Samples topics discovered using C-NTM for large scale datasets.

The proposed model can be extend in many directions. The continuous document embedding model is based on a simple single-hidden-layer auto-encoder. The use of more sophisticated models such as variational auto-encoders such as proposed by Kingma & Welling (2013) could be investigated. In the direction of using more sophisticated neural networks, proposed probabilities of words given topics and topics given documents models could be represented with deeper neural networks which are known to have high representation power. This could lead to decisive improvements, specially for large scale corpora.

Another possible direction would be to integrate proposed neural topic models into models combining probabilistic topic models and neural network model such as done by Dieng et al. (2017) who combines LDA to capture global words semantic to recurrent neural network language models. This would allow design a model into a single neural network framework. The model would be fully trainable with stochastic gradient descent on sample batches.

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
