# OpenReview forum: "Discovering Topics With Neural Topic Models Built From PLSA Loss"
_ICLR.cc/2020/Conference — Reject_

### Official Review · AnonReviewer1 · 2019-10-21
**Official Blind Review #1**

**Rating:** 1

**Review:**

First, some minor issues.  I didn't understand equation (3).  It seems to be a variant of equation (4), and seems to be in disagreement with equation (6).  Might be better if the equation was just dropped.  For equation (9), you should have brackets "()" around the argument to the exp.

Second, in terms of comparisons, the paper lacks adequate related work.  Some non-parametric but non-neural
models not implemented in GPUs substantially beat LDA, and will run on all the big data sets you list, though perhaps
not quickly!   There has also been a number of neural and hybrid topic models developed.
DocNADE and LLA (Zaheer etal), for instance, work very well in PPL.  Then there are many new deep topic models.  Some use the amortised inference that you adopt in section 5.    Some incorporate word embeddings or document metadata to
further improve performance metrics.  Note some of the earlier ICLR/NeurIPS papers with deep models didn't
do extensive comparative empirical testing, so may not work well against DocNADE or more recent algorithms.

In terms of related work, topic models is a bit of a mine-field because there is a huge amount of work in
a huge number of venues, and few authors do a good job of covering related work.  What you have listed are mainly the
older works.  Recent work also includes Poisson Matrix Factorisation and its variants, as well as hierarchical
variants of LDA, much better than the 2004 paper you mention.

To do the coherence comparisons, easiest way is to use the Palmetto software.
You can also evaluate models by using them as features in a classification task.

It was interesting that you only did one layer for your networks, i.e.,  equations (4)-(6).  Why was this?
I would have liked to have seen the impact of more layers. However, your model is remarkably simple
so if it works well, that is good.

Anyway, the experimental evaluation shows good results on all three datasets for your models, but its hard to be sure
since you only have one comparison, an old LDA, and nothing recent.  So promising work, but
related work and experimental work need to be improved.



**Experience Assessment:**

I have published in this field for several years.

**Review Assessment: Checking Correctness Of Derivations And Theory:**

I carefully checked the derivations and theory.

**Review Assessment: Checking Correctness Of Experiments:**

I carefully checked the experiments.

**Review Assessment: Thoroughness In Paper Reading:**

I read the paper thoroughly.

---

### Official Review · AnonReviewer2 · 2019-10-23
**Official Blind Review #2**

**Rating:** 1

**Review:**

This paper proposes a neural topic model that aim to discover topics by minimizing a version of the PLSA loss. According to PLSA, a document is presented as a mixture of topics, while a topic is a probability distribution over words, with documents and words assumed independent given topics. Thanks to this assumption, each of these probability distributions (word|topic, topic|document, and word|document) can essentially be expressed as a matrix multiplication of the other two, and EM is usually adopted for the optimization. This paper proposes to embed these relationships in a neural network and then optimize the model using SGD.

I believe the paper should be rejected because: 1) most aspects of this paper are a little dated 2) novelty is little 3) experimental section is very limited and unconvincing.

To elaborate on the experimental section:
- Only LDA has been presented as baseline. There's plenty of neural topic models to compare against (you mentioned some in your related work section) but no comparison with any of those is presented. If the concern is their training time on large datasets, they should be at least presented as comparison for the smaller datasets. For the large datasets there's other approaches that would scale and should be presented as baselines: 1) train on a sample of the dataset 2) co-occurrence based topic methods on sliding windows of text are extremely fast (eg see "A Biterm Topic Model", "A Practical Algorithm for Topic Modeling with Provable Guarantees", and "A Reduction for Efficient LDA Topic Reconstruction" which could fit your scenario with large datasets where topics most likely have small overlap with each other and are almost separable by anchor words.)
- Even regarding just LDA: what hyper-parameters \alpha and \beta did you set for LDA? Tuning \beta to a small value might have an impact for large datasets.
- Metrics: only perplexity is presented and metrics but it's well known that perplexity on its own is quite limited and often is not correlated to human judgment. Consider adding topic coherence measures as well.
- The section on continuous document embeddings is confusing and the explanation should be improved and the formalism tightened.


Other (did not impact the score):
- Biases: you're adding biases to your probability estimation equations. This is not in line  with the PLSA assumption. What happens if no biases are used?

The paper has several typos and grammatical errors, e.g.:
- page 2, L#1: networks -> network
- page 4, sec 3.2: set unobserved -> set of unobserved
- page 5, sec 5: pratise -> practice
- several places: it's -> its


**Experience Assessment:**

I have published one or two papers in this area.

**Review Assessment: Checking Correctness Of Derivations And Theory:**

N/A

**Review Assessment: Checking Correctness Of Experiments:**

I assessed the sensibility of the experiments.

**Review Assessment: Thoroughness In Paper Reading:**

I read the paper at least twice and used my best judgement in assessing the paper.

---

### Official Review · AnonReviewer3 · 2019-10-30
**Official Blind Review #3**

**Rating:** 3

**Review:**

I am unimpressed with the quality of writing and presentation, to begin with. There are numerous grammatical errors and typos that make the paper a very difficult read. The presentation also follows an inequitable pattern where the backgrounds and related works are overemphasized and the actual contribution of the paper seems very limited. In its current form, this paper is not ready for publication in ICLR.

The idea of representing a document as an average of the embeddings of the words is a rather crude idea. Paragraph2vec and many of its derivatives have shown significant improvements with document modelling. The perplexity improvements are nice to have, but I would have liked to see the embeddings being applied to some supervised problems to assess their utilities.

There are quite a few computationally expensive normalization terms. I am curious to understand how these summations do not slow the training process down without further approximations. The authors may present some computational complexity measures to convince readers about the practical applications of the proposed models.


**Experience Assessment:**

I have published one or two papers in this area.

**Review Assessment: Checking Correctness Of Derivations And Theory:**

I assessed the sensibility of the derivations and theory.

**Review Assessment: Checking Correctness Of Experiments:**

I assessed the sensibility of the experiments.

**Review Assessment: Thoroughness In Paper Reading:**

I read the paper at least twice and used my best judgement in assessing the paper.

---

### Public Comment · ~pankaj_gupta1 · 2019-09-27
**Missing References, missing comparisons with recent Neural topic models, incomplete evaluation**

Following are the missing references, especially in Neural topic modeling:

[1] Hugo Larochelle and Stanislas Lauly. A neural autoregressive topic model. In NIPS 2012.
[2] Pankaj Gupta, Yatin Chaudhary, Florian Buettner, and Hinrich Schuetze. Document informed neural autoregressive topic models with distributional prior. In AAAI 2019.
[3] Pankaj Gupta, Yatin Chaudhary, Florian Buettner, and Hinrich Schuetze. textTOvec: Deep Contextualized Neural Autoregressive Topic Models of Language with Distributed Compositional Prior. In ICLR 2019.
[4] Akash Srivastava and Charles Sutton. Autoencoding variational inference for topic models. In  ICLR 2017.

Please include the reference [3] for the mentions of combining topic and language models (e.g. in conclusion).

Additional Comments:
1. Why are the perplexity values are too high?
2. Please include a quantitative comparison with other neural topic models [e.g., 1, 2, 3, 4].
3. What do the high perplexity scores signify?
4. To better demonstrate the applicability of topic models, could you include additional evaluation such as topic coherence for quality of topics, document clustering or classification or retrieval, similar to [2, 3, 4]?

---

> ### Author Response · Authors · 2019-09-30
> **Adding references, experiments and comparison**
>
> Thanks for your feedbacks Pankaj. They will be taken into accounts in the coming days. I will come back to you as soon as they are done.

---

> ### Author Response · Authors · 2019-10-07
> **Responses to Pankaj Gupta about missing references, comparisons, and evaluation**
>
> Dear Pankaj
> Again thank you for your feedbacks on our paper.  Here we respond to your concerns.
> First we accounted about missing reference you mentionned and added them to the paper. We note that Larochelle & Lauly was already cited in the related work section.
>
> About your question about related to the perplexity that are high. This is due to the fact that our vocabulary are not filtered: we used all the words appearing in the document. Just to show that, we designed an experiment on TwentyNewsGroup dataset where we used as vocabulary word appearing more than: 20, 40, 60, 80, and 100 times. These results will be added to the paper. When using words appearing more than 100, perplexity are much lower. But this did not change any conclusions.
>
> About your concerns related to coherence scores, we added results about UMAss coherence scores (Mimno et al Optimizing semantic coherence in topic models. EMNLP 2011).
>
> About your concerns related to comparison with neural topics models, some comparisons with such methods will be added to the paper. In the first version we compared mainly to LDA because it remains the most popular unsupervised topic model.
>
> We will also display TSNE based document embedding for the TwentyNewGroupDataset which show that documents cluster according to their categories
>
> Hope these responses answer your concerns.

---

### Decision · Program_Chairs · 2019-12-19

**Decision:**

Reject

**Comment:**

This paper presents a neural topic model with the goal of improving topic discovery with a PLSA loss. Reviewers point out major limitations including the following:

1) Empirical comparison is done only with LDA when there are many newer models that perform much better.
2) Related work section is incomplete, especially for the newer models.
3) Writing is unclear in many parts of the paper.

For these reasons, I recommend that the authors make major improvements to the paper before resubmitting to another venue.